# How Do Patients with Life-Limiting Illness and Caregivers Want End-Of-Life Prognostic Information Delivered? A Pilot Study

**DOI:** 10.3390/healthcare9070784

**Published:** 2021-06-22

**Authors:** Ebony T. Lewis, Kathrine A. Hammill, Maree Ticehurst, Robin M. Turner, Sally Greenaway, Ken Hillman, Joan Carlini, Magnolia Cardona

**Affiliations:** 1School of Population Health, The University of New South Wales, Sydney 2052, Australia; 2School of Psychology, The University of New South Wales, Sydney 2052, Australia; 3School of Science and Health, Western Sydney University, Campbelltown 2560, Australia; K.Hammill@westernsydney.edu.au; 4South Western Sydney Clinical School, University of New South Wales, Liverpool 2170, Australia; mareeticehurst1@optusnet.com.au (M.T.); k.hillman@unsw.edu.au (K.H.); 5Biostatistics Unit, Otago Medical School, University of Otago, Dunedin 9054, New Zealand; robin.turner@otago.ac.nz; 6Supportive and Palliative Medicine, Westmead Hospital, Westmead 2145, Australia; Sally.Greenaway@health.nsw.gov.au; 7Intensive Care Unit, Liverpool Hospital, Liverpool 2170, Australia; 8Department of Marketing, Griffith University, Southport 4222, Australia; j.carlini@griffith.edu.au; 9Institute for Evidence Based Healthcare, Bond University, Robina 4226, Australia; mcardona@bond.edu.au; 10EBP Professorial Unit, Gold Coast University Hospital, Southport 4215, Australia

**Keywords:** terminally ill, older adults, prognosis, patient preference, decision making

## Abstract

We aimed to identify the level of prognostic disclosure, type of prognostic information and delivery format of prognostic communication that older adults diagnosed with a life-limiting illness or caregivers prefer to receive. We developed and pilot tested an open-ended survey to 15 older patients and caregivers who had experience in health services for life-limiting illness either for a relative, friend or themselves. Five hypothetical clinical scenarios of prognostic options were presented to ascertain preferences. The preferred format to receive prognostic information was verbal delivery by the clinician with a written summary. Photos and videos were less favoured, and a table with numbers/percentages was least preferred. Distress levels to the prognostic scenarios were low, with the exception of a photo. We conclude that older patients/caregivers want end-of-life prognostic information delivered the traditional way, verbally by clinicians. Options to deliver prognostic information may vary across patient groups but empower clinicians in introducing end-of-life discussions with patients/caregivers. Our study illustrates the feasibility of involving terminal patients and caregivers in research that contributes to eliciting prognostic preferences. Further research is needed to understand whether the prognostic preferences of hospitalized patients with life-limiting illness differ.

## 1. Introduction

Medicine is moving away from a paternalistic approach towards shared decision making [1], often involving discussions about prognosis with the patient and their caregivers [2,3]. Shared decision making (SDM) can be defined as a “consultation process where a clinician and patient jointly participate in making a health decision, having discussed the options and their benefits and harms, and having considered the patient’s values, preferences and circumstances” [4]. Many express a desire for truth disclosure when diagnosed with a life-limiting illness [5,6,7]. However, the level of prognostic information people want disclosed by their clinician varies [8], and not all patients want to participate in decision making [9], instead choosing to take a more passive role in treatment decision making [10]. This becomes even more challenging for dementia patients [11], where family members and caregivers are the recipients of diagnostic and prognostic information from the treating clinician and act as the default decision makers.

Prognostic disclosure can help facilitate appropriate goals of care [12] in older adults who have life expectancy of less than 10 years and the very old (85+ years) [13]. However, clinicians may avoid discussing prognosis with adults with a life-limiting illness [14] and often find these conversations difficult due to prognostic uncertainty about time to death [14,15,16], desire to preserve patient hope and cultural factors [15,17].

A variety of techniques have been recommended and successfully implemented to deliver prognostic information in the cancer realm [18]. However, there is a paucity of literature discussing the best way to deliver prognostic information to older adults diagnosed with life-limiting nonmalignant diseases and their caregivers. For this reason, a study was undertaken to develop and test the feasibility of the recruitment of older people, the administration of a new survey instrument and the enactment of a preliminary assessment of the level of disclosure, type of prognostic information and delivery format that older adults diagnosed with life-limiting illness or their caregivers prefer to receive, in order to confirm the appropriateness of the collected data.

## 2. Materials and Methods

### 2.1. Participants

The first fifteen people who were available from a University of New South Wales (UNSW) consumer group comprised of 37 adults were selected as participants. The consumer group eligibility included adults who have direct experience in health services for life-limiting illness, including terminal care either for a relative, friend or themselves; experience in providing physical and social aspects of care for terminal older adults and/or their family members or a commitment to the concept of improving the end-of-life experience for themselves or others. They were recruited through a network advertisement by email. The study had ethics approval from UNSW (# HC16159).

### 2.2. Design

An open-ended survey questionnaire was developed by the study team based on the study objectives with input provided by practising clinicians, including an intensivist, palliative care physician and aged care and community nurses, to ensure that the hypothetical scenarios reflected the Australian clinical environment. Data were collected between June and July 2017 in Sydney, Australia, either face to face or over the phone via the survey containing closed- and open-ended questions (Appendix A), visual media, and Likert scales. For phone interviews, visual media was emailed to the participant just prior to interview. Participants were prompted by the researchers (EL and MT) during the interview to view five hypothetical clinical scenarios depicting how they would prefer health information be communicated: (i) verbal with written summary; (ii) table format; (iii) photo; (iv) video; (v) graph (Appendix A). A 5-point Likert scale was used for each scenario for their preferred format and how distressing each format was. Respondents had the opportunity to expand on reasons for their preference or dislike of each scenario. Prior to the interview, the scenario sequence was randomly assigned by an external statistician using a computer-generated list of random number sets. Interviewers administered the random sets following the list in consecutive order. Respondents could elect to answer for themselves or on the behalf of the person they cared for.

### 2.3. Data Analysis

The narrative description of the process, outputs supplemented with qualitative data transcribed verbatim and managed in Nvivo version 11 and survey data were entered into IBM SPSS Version 24. Two researchers (ETL and KAH) independently conducted a thematic analysis of the data and later compared findings and inferences.

## 3. Results

### 3.1. Participant Characteristics and Process

Fifteen health service consumers (12 women and 3 men) participated in the pilot study, with over a third (5) having been born outside of Australia and the majority (12) acting in the role of a ‘caregiver’ (Table 1). All participants completed all study procedures without impediments, from consent to full response to all questions. All interviews took place at scheduled times. Despite interviewers not being physically present, recruitment by email was uneventful, and there was no ambiguity in relation to any aspect of the telephone interview format or understanding of the questions; and no complaints were made about the interview duration, privacy or security and no adverse events were reported.

### 3.2. Level and Type of Prognostic Disclosure

All participants (15/15) reported wanting full prognostic information disclosed by the treating team. Respondents stated various reasons for wanting full prognostic information. For most, it was their need for control over their situation. “If I’m to make any decisions I need as much info as possible…” [Terminally ill patient, Female, 70–79 years]. The provision of information enabled caregivers and patients to know “what to expect” [Caregiver, Female, < 70 years] in relation to signs and symptoms as conditions progressed. This knowledge was thought to enhance the respondent’s ability to prepare for the future and to understand “what my limits are” [Terminally ill patient, Male, 70–79 years], and if there was “anything else [they] could do” [Caregiver, Female, < 70 years]. For caregivers, being armed with all of the required information was perceived to equip them with the knowledge to adequately care for their loved one: “Don’t think as a carer I can do the job without all information” [Caregiver, Female, < 70 years]. However, there was variation between what prognostic information each required (Table 2). Over half the participants sought information on expected survival time, followed by treatment benefits and alternatives. The impact of treatment, including the harm and side effects and impact on the person’s quality of life, was also frequently expressed.

### 3.3. Delivery Format of Prognostic Information

All participants stated they would prefer to receive prognostic information verbally by the treating clinician and would be least likely to prefer receiving information pictorially (Table 3). In order of priority, the preferred way in which health information was communicated in each clinical scenario was verbal delivery in conjunction with a written summary (Appendix A), as “it has a personal touch” [Caregiver, Female, <70 years]. Participants least preferred to receive prognostic information in table format, as they perceived it to be “hard to understand and everyone is different” [Caregiver, Female, <70 years].

Participants also made other suggestions for how they would like prognostic information delivered, predominantly in the form of trusted written resources they could access on their own time (Table 3).

### 3.4. Impact of Format of Health Information Communicated

The highest levels of distress occurred in the picture and video scenarios (Appendix A). Participants found the picture of a person resting in an ICU bed “too confronting” [Caregiver, Female, <70 years] as “images can stay in your mind” [Caregiver, Female, 70–79 years], while the video of a patient receiving CPR was perceived as particularly distressing as “you are reminded of what you may have to go through” [Terminally ill patient, Female, 70–79 years]. Conversely, participant distress levels were lowest when presented with the verbal scenario with the written summary, with 12/15 stating no distress at all.

## 4. Discussion

This pilot study confirmed the feasibility of recruiting and the acceptability of interviewing older patients and caregivers on sensitive topics, such as confronting bad news and choosing their delivery mode. Despite clinicians’ reluctance to communicate with older patients and their caregivers about poor prognosis [15], our study shows that it is feasible to involve community-based terminally ill patients and their caregivers in research that contributes to eliciting information about their prognostic preferences. Participants were accepting of the questions and scenarios by displaying little or no distress and not withdrawing during the interviews. A previous attempt to raise sensitive conversations using vignette techniques in Australia yielded a high acceptability and low level of missing data [19]. Another recent pilot in Norway also demonstrated the feasibility of initiating difficult conversations on advance care planning with hospitalized patients [20].

Our study also added new practical information to the literature about the type of prognostic information, the degree of disclosure and the formats in which this information is delivered to older community-dwelling people. It found that these participants want full prognostic information disclosed, and information on survival time is highly desired, with the preferred format of receiving prognostic information to be verbal information from the clinician accompanied by a short written summary. This finding on the preference for a doctor to deliver bad news has been reported before in Asia [21], suggesting there is an issue of trust in the clinician’s expertise.

Our preliminary findings are consistent with previous reports that older community dwellers and caregivers favour life expectancy discussions when the patient’s survival time is shorter [8]. Our results also have similarities with an earlier Australian study finding that a large proportion of middle-aged women with breast cancer wanted verbal prognostic information summarized in written format, where words were preferred over numbers [22]. Contrary to our study, a recent study in patients with end-stage renal disease found the use of graphical formats for survival outcomes to be highly favoured and useful [23], and another inpatient study of >75-year-olds reported that the pictorial representation of probabilities was positively received [24]. Cultural differences and education may explain this variation in preferences, but our pilot sample did not intend and did not allow for statistical exploration of these factors. Education has previously been associated with patients wanting high levels of prognostic information [25]. A systematic review on the content, style and timing of information found that as the life-limiting illness trajectory progressed, patients wanted less information about prognosis, while caregivers sought more information [26]. Another community survey in the USA reported that the avoidance of health information was a common coping mechanism to maintain hope [27]. Most of our respondents were caregivers and educated at least at a secondary level, which may explain the differences.

The clinician’s ethical duty to break bad news and allow time for patients and families to come to terms with the inevitable needs to be balanced with the patient readiness for hearing poor prognosis [28]. The preference to receive full prognostic disclosure varies across cultures and settings and may be changing with time, as suggested by the findings of two population surveys in 2015 in Brazil [29] and Australia [30] reporting that a majority of older members of the public wanted to know if they had less than a year to live. By contrast, a 2003 survey of older community dwellers with life-limiting illness and their caregivers in the USA [31] and a 2008 Canadian survey of inpatients with advanced chronic lung disease [32] reported that many did not want prognostic disclosure.

Our ambulatory patients and caregivers reported more distress with the visual mediums than with written information, which contrasts with a recent study of older inpatients with an expected survival status of < 1 year, where only a minority felt uncomfortable watching videos on CPR and intubation simulation [33]. Replication of this preliminary survey is intended in a hospital setting to uncover other possible variations in preferences. There may still be merit in the use of visual aids, as they can also help those with low health and/or numeracy literacy, language barriers and cognition factors [34], and it is known that clinicians see benefit in supplementing their verbal prognostic communication with an appropriate visual resource [35].

This pilot test of a survey instrument under development covered the opinions of noncancer patients with a life-limiting illness or their caregivers, exposing them to a random sequence of scenarios to elicit preferences. While our participants were English speaking, they represented several ethnic groups, with a third of our sample born outside of Australia. Our study has some limitations: it was overly represented by caregivers, with the majority being highly educated females, but this reflects a sociodemographic reality among the chronically ill in our catchment area. Gender differences in prognostic disclosure in advanced cancer patients have been previously observed by others, with men more likely than women to want prognostic information [36], and prognostic preferences between patients and caregivers have also been found to differ [26]. As ours was a pilot study for the purpose of testing a questionnaire and clarifying the data useability, we selected a small sample, as customary in pilots, and did not examine these differentials. This will be incorporated in the results of our larger quantitative study, which is underway. Whilst examining cultural differences was beyond the scope of this study, further research is warranted to investigate the impact that culture and ethnicity may play in the level of disclosure, type of prognostic information and delivery format which terminally ill older patients and caregivers prefer to receive. The preferences of older people with life-limiting illness admitted to hospital as inpatients or aged care residents may vary substantially from those of community dwellers, so our results are not generalizable. In clinical practice, having options to deliver prognostic information may assist with the introduction of end-of-life discussions with patients/caregivers and empower both clinicians and patients to plan and agree on goals of care.

## 5. Conclusions

It is feasible and acceptable to interview older community-based patients and caregivers about their prognostic preferences. Our findings suggest they want end-of-life prognostic information delivered in a verbal format with a short written summary and prefer full prognostic disclosure from clinicians, with life expectancy and treatment benefits. The use of graphs, videos, pictures or numeric tables was not welcomed by the majority. Further research is needed to understand the preferences of hospitalized older adults near the end of life and their caregivers.

## Figures and Tables

**Table 1 healthcare-09-00784-t001:** Respondent characteristics.

Parameter	n
Age of respondent (years)	
<70	8
70–79	6
80+	1
Female	12
Respondent role	
Caregiver	12
Self	3
Education level	
Primary	0
Secondary	2
Trade/technical	5
Tertiary	8
Aboriginal and/or Torres Strait Islander	1
Country of birth	
Australia	10
Outside of Australia	5

**Table 2 healthcare-09-00784-t002:** Type of prognostic news respondents are interested in learning from their health service provider.

Prognostic Issue †	n
Expected survival time	8
Benefits of treatment	7
Results with other treatment alternatives	7
Treatment harms or side effects	6
Treatment impact on quality of life	6
Impact of management on caregivers	4
Disease trajectory	4
Complications in the next six months	2
Costs of treatment	2
Symptom management	2
Chances of cure with treatment	1
Option for non-medical treatments	1

† Types of prognostic issues were grouped from an open-ended survey question.

**Table 3 healthcare-09-00784-t003:** Delivery format of prognostic information from clinician.

Format	N
Verbal information from the doctor †	15
Some graph which shows good and bad consequences ^†^	5
Video with supported explanation of treatment ^†^	4
Table with number of probabilities based on previous studies ^†^	4
Picture of patient undergoing a treatment ^†^	3
Other Suggestions	
Written information such as pamphlets and books	3
Internet and Google links	2
Real-life stories with people experiencing similar situation	2
Research studies	2
Flow charts	1

^†^ Respondents could choose more than one format category.

## Data Availability

Not applicable.

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
