# Peer review of "How Do Patients with Life-Limiting Illness and Caregivers Want End-Of-Life Prognostic Information Delivered? A Pilot Study"

_healthcare, 2021, doi:10.3390/healthcare9070784_

Round 1
Reviewer 1 Report
The manuscript is focused on the important clinical dilemma which needs to be addressed, more researched and introduced to the clinicians all over the world: the potential implementation of common system of prognostic information delivery to a terminally ill patient.
The research touches quite a sensitive ethical issue, and I support the importance of the topic, which has been covered in this manuscript.
I totally agree with the statement on a necessity for more to be done to improve the quality of the last precious moments for the patient’s survival and peace of mind, and their special need for a compassionate and loyal conversation, as well as presence of the full disclosure of the diagnostics results, which can help to make a correct final decision.
However, I would recommend, for your future consideration, to increase the number of patients tested in your research and make more precise data validation based on the analysis of the larger pool of the man participants in the individuals’ groups.
In general, I support the research results stating that support delivery end-of life prognostic information in verbal format with a short-written summary combined with a full prognostic disclosure from clinicians, is important for longer life expectancy and a potential treatment benefits'.
Reviewer 2 Report
In this manuscript, the authors have interviewed older adults who had direct experience in health services for life-limiting illness to identify the information they prefer to receive about the prognosis, risks, and choices. Also, the interview addresses the format of communication that is preferred and exerts a lesser level of distress.
The objectives of this pilot study are clear and interesting since patients have the right of making informed decisions about their health. Also, caregivers often need to know about the illness to better approach the care and even their own self-care. Thus, the research is of interest. Also, the best option for all the participants is to have direct information from the doctor. This result is clear, and it generates no doubts. However, the authors should address some considerations.
My major concern with the study conducted here is the sample and the management of the information analysed. Of 15 people, 12 are women, and only 3 are men, and also 12 are caregivers, and 3 are patients. Although it is a pilot qualitative study, it would be desirable to include a comparable number of men and women and patients and caregivers in the inclusion criteria. Furthermore, given that the methodology seems quite affordable (an interview), I would strongly recommend including a greater number of participants in the study. However, I understand that current times are really hard for research, even more in a study including older people. So, if it is not possible to increase the sample size, the data analysis and the discussion should be improved.
Also, the results should address these questions: are they any sex difference in the preferences for health information? Also, the perspective of caregivers might be different from that of patients. Unfortunately, these considerations have not been included in the results or even discussed. The authors also suggest possible cultural differences, but they are not reflected in the study. In case there is impossible to include more participants in the study (which could increase the robustness of the study), at least the authors should approach these questions.
Some other minor comments:
- In the title and the text, it would be better to refer to "patients" and "caregivers" instead of "customers"
- abstract and 53: Revise the usage of ";"
- the inclusion and exclusion criteria should be explicitly stated
- the authors are encouraged to include in the text or supplementary material the questions included in the interview. Which were the questions? Were they open or closed questions?
- Table 2: These options were addressed utilizing open questions or as options? In case there were open questions, it is reasonable to interpret the data like positive responses. Still, if there were options, the percentages are very low and can be interpreted as the opposite... About half of the participants are not interested in learning about expected survival time, benefits of treatment, alternatives, side effects... And the majority did not report to be interested in learning about the impact on caregivers, disease trajectory, complications, costs, symptom management or chances of cure... It needs to be further discussed.
- The limitations of the study should be clearly stated.
